# A Systematic Review to Compare Adverse Pregnancy Outcomes in Women with Pregestational Diabetes and Gestational Diabetes

**DOI:** 10.3390/ijerph191710846

**Published:** 2022-08-31

**Authors:** Nompumelelo Malaza, Matladi Masete, Sumaiya Adam, Stephanie Dias, Thembeka Nyawo, Carmen Pheiffer

**Affiliations:** 1Biomedical Research and Innovation Platform (BRIP), South African Medical Research Council, Tygerberg, Cape Town 7505, South Africa; 2Department of Obstetrics and Gynaecology, Faculty of Health Sciences, University of Pretoria, Pretoria 0001, South Africa; 3Diabetes Research Centre, Faculty of Health Sciences, University of Pretoria, Pretoria 0001, South Africa; 4Centre for Cardio-Metabolic Research in Africa, Division of Medical Physiology, Faculty of Medicine and Health Sciences, Stellenbosch University, Tygerberg, Cape Town 7505, South Africa

**Keywords:** type 1 diabetes mellitus, type 2 diabetes mellitus, gestational diabetes mellitus, adverse outcomes, pregnancy

## Abstract

Pregestational type 1 (T1DM) and type 2 (T2DM) diabetes mellitus and gestational diabetes mellitus (GDM) are associated with increased rates of adverse maternal and neonatal outcomes. Adverse outcomes are more common in women with pregestational diabetes compared to GDM; although, conflicting results have been reported. This systematic review aims to summarise and synthesise studies that have compared adverse pregnancy outcomes in pregnancies complicated by pregestational diabetes and GDM. Three databases, Pubmed, EBSCOhost and Scopus were searched to identify studies that compared adverse outcomes in pregnancies complicated by pregestational T1DM and T2DM, and GDM. A total of 20 studies met the inclusion criteria and are included in this systematic review. Thirteen pregnancy outcomes including caesarean section, preterm birth, congenital anomalies, pre-eclampsia, neonatal hypoglycaemia, macrosomia, neonatal intensive care unit admission, stillbirth, Apgar score, large for gestational age, induction of labour, respiratory distress syndrome and miscarriages were compared. Findings from this review confirm that pregestational diabetes is associated with more frequent pregnancy complications than GDM. Taken together, this review highlights the risks posed by all types of maternal diabetes and the need to improve care and educate women on the importance of maintaining optimal glycaemic control to mitigate these risks.

## 1. Introduction

Globally, it is estimated that 21.1 million (16.7%) live births in 2021 were associated with maternal diabetes [1]. Of these, 10.6% were due to pregestational type 1 (T1DM) and type 2 (T2DM) diabetes mellitus, 9.1% were due to T1DM or T2DM first detected in pregnancy and 80.3% were due to gestational diabetes mellitus (GDM), a milder form of hyperglycaemia that develops in the second trimester [1]. Normal pregnancy is characterised by insulin resistance and requires an increased pancreatic β-cell response in order to maintain normoglycaemia [2]. GDM develops in women who are unable to mount a compensatory β-cells response, leading to hyperglycaemia. Increasing maternal age, along with increasing rates of obesity and diabetes worldwide, has led to rising rates of diabetes in pregnancy [1,3,4]. Obesity has been identified as a significant risk factor for maternal diabetes. A meta-analysis of 20 studies reported that women who were overweight (2.1-fold), obese (3.6-fold) or severely obese (8.6-fold) had a higher risk of developing diabetes compared to normal-weight pregnant women [5].

Maternal diabetes is associated with pregnancy complications and increased rates of adverse maternal and neonatal outcomes [6,7]. Short-term complications include macrosomia, large for gestational age (LGA), respiratory distress syndrome (RDS), neonatal hypoglycaemia, neonatal intensive care unit (NICU) admission, intrauterine growth restriction, congenital anomalies, preterm birth, pre-eclampsia, caesarean section (C/S) and preterm birth while in the long-term both mothers and their babies have an increased risk of metabolic disease [8,9,10]. Women with GDM have a ~7-fold increased risk of developing T2DM [11] and a ~4-fold increased risk of developing cardiovascular and coronary artery disease after pregnancy [12], while pregestational diabetes predisposes women to developing diabetes-related complications such as retinopathy and nephropathy or may accelerate the course of these complications if they already exist [4,7,13].

It is widely reported that all types of maternal diabetes are associated with pregnancy complications; although, adverse outcomes are more common in women with pregestational diabetes [14,15,16,17,18]. As adverse pregnancy outcomes are closely related to poor glycaemic control and the first trimester being a critical period for organogenesis, it is speculated that preconception hyperglycaemia and the longer time of exposure to hyperglycaemia in utero may contribute to the complications associated with pregestational diabetes [19].

Despite the large body of evidence that associates pregestational diabetes with more frequent adverse pregnancy outcomes than GDM [20,21,22,23,24,25], conflicting results have been reported [17,22,26,27,28]. This review aims to summarise and synthesise studies that have compared adverse pregnancy outcomes in pregnancies complicated by pregestational diabetes and GDM. Three databases, Pubmed, Scopus and EBSCOhost were searched to identify eligible studies, which were summarised and synthesised using systematic review methods. Commonly reported adverse pregnancy outcomes in literature [29] were selected for inclusion in this review. These include congenital anomalies, pre-eclampsia, neonatal hypoglycaemia, macrosomia, NICU admission, stillbirth, Apgar score, large for gestational age (LGA), induction of labour (IOL), respiratory distress syndrome (RDS) and miscarriages.

## 2. Methodology

This systematic review was conducted adhering to the Preferred Reporting Items for Systematic Reviews and Meta-Analyses (PRISMA) guidelines [30] (Appendix A).

### 2.1. Search Strategy and Study Selection

Three databases, Pubmed, Scopus and EBSCOhost were searched for studies reporting on maternal diabetes and pregnancy outcomes, published between January 1993 and December 2021. The search terms included “type 1 diabetes mellitus” or “type 1 diabetes” or “diabetes mellitus type 1” or “diabetes type 1” and “type 2 diabetes” or “type 2 diabetes mellitus” and “pre-gestational diabetes” or “gestational diabetes” or “diabetes in pregnancy” and “pregnancy complications” or “perinatal outcomes” or “adverse outcomes” or “pregnancy outcomes” and were adapted to each database. An experienced information scientist was consulted to ensure that the search terms were relevant and optimally arranged. References were managed in Zotero 5.0.96.2 (Corporation for Digital Scholarship, Vienna, VA, USA). After the removal of duplicate studies, two reviewers (NM and MM) independently screened articles for eligibility. Disagreements or uncertainties were resolved by discussion and consensus or in consultation with a third reviewer (CP). Additionally, references from selected articles were screened for potentially relevant articles.

### 2.2. Inclusion and Exclusion Criteria

Studies that compared pregnancy outcomes in one or two types of maternal diabetes only, those focusing on other forms of diabetes (maternal onset of diabetes in young (MODY), etc.), abstracts, review articles, letters, case reports, intervention studies and those not written in English, were excluded. Review articles were screened to identify eligible studies that may have been missed using our search strategy. Studies reporting on adverse outcomes in pregnancies complicated by T1DM, T2DM and GDM were included. This systematic review was conducted to answer the following question:Is there an association between maternal diabetes type and the frequency of adverse pregnancy outcomes?

This was achieved using the following:Participants—Pregnant women with GDM;Intervention—No intervention was used in this study;Comparator—Pregnant women with pregestational T1DM and T2DM;Outcome—Pregnancy outcomes.

### 2.3. Data Extraction and Quality Assessment

Data that were extracted and recorded included author details (name and date of publication), study details (aim and design, study period and GDM diagnostic criteria), sample size, characteristics of the population (ethnicity), country and pregnancy outcomes in the different diabetic groups. Two reviewers (NM and MM) independently appraised the study quality and risk of bias using the Newcastle–Ottawa Scale. The Newcastle–Ottawa Scale is used to assess the quality of non-randomized studies, such as case-control and cohort studies [31]. It assesses study quality based on three study parameters: selection, comparability, and outcomes, which are divided into eight specific items that can be scored as one or two points with points totalling nine (Appendix A). Disagreements between the two reviewers were resolved by consulting a third reviewer (CP). A study was classified as having a low risk of bias (7 to 9), moderate (5 to 6) or high risk of bias (1 to 4) based on the total score.

### 2.4. Definitions of Pregnancy Outcomes

Caesarean section refers to the delivery of a foetus through an incision in the abdominal wall and uterus [32]. Preterm birth is defined as birth before 37 weeks of gestation [33]. Congenital anomalies are defined as structural or functional anomalies that occur during intrauterine life as determined by the ultrasound scan and laboratory tests [23]. Preeclampsia is defined as hypertension (>140/90 mm Hg) and proteinuria (>0.3 g of protein in a 24 h urine collection) developing after 20 weeks of gestation [34]. Macrosomia is defined as giving birth to babies weighing > 4000 g [29]. Stillbirth is foetal death after 24 weeks of gestation or foetus > 500 g [35]. LGA is defined as birth weight > 90th percentile for age [36]. Neonatal hypoglycaemia is defined as a plasma glucose value <1.65 mmol/L in the first 24 h of life and <2.5 mmol/L onwards [37]. NICU admission refers to the admission of a newborn to an intensive care unit for specialised care due to a critical condition or illness [38]. Miscarriage refers to foetal death before 24 weeks of gestation or foetus < 500 g [39]. Induction of labour refers to the process that involves mechanical or surgical means to initiate uterine contractions [40]. The Apgar score is used to assess the well-being of a neonate at 1 min and 5 min after birth [41]. Respiratory distress syndrome is defined as the need to supplement oxygen to the neonate to maintain a saturation over 85% within the first 24 h after birth [42].

## 3. Results

### 3.1. Selected Studies

A total of 2164 studies were identified from the search strategy. An additional three articles were identified by reviewing the reference lists of relevant articles and reviews resulting in 2167 articles. After removing duplicates, 1958 article titles and abstracts were screened for eligible full-text articles. We excluded studies that compared one or two types of maternal diabetes only, interventional studies, those not written in English, review articles, letters, case reports and abstracts. A total of 20 studies, published between January 1993 and December 2021, met the inclusion criteria and are discussed in this review (Figure 1).

### 3.2. Characteristics of Included Studies

Twenty articles published between 1993 and 2021 were included in the review (*n* = 196,232 participants; Appendix A). These studies were conducted across five continents (Europe, Asia, North America, Africa and Australia). Sixteen studies were retrospective, two were prospective, one was cross-sectional and one was unspecified. Nine studies reported adverse outcomes for pregestational diabetes, combining data for T1DM and T2DM [16,20,22,25,27,43,44,45,46], while 11 studies reported data for T1DM and T2DM, separately [14,15,17,21,24,47,48,49,50,51,52]. These studies reported on various maternal and neonatal short-term pregnancy adverse outcomes, of which 13 are summarised in this review. These selected adverse outcomes are amongst the most common in the literature [29]. None of the studies investigated long-term maternal outcomes in women with T1DM, T2DM and GDM.

The studies in this review used different diagnostic criteria for GDM, which included the International Association of Diabetes and Pregnancy Study Groups Consensus Panel, 2010 (IADPSG; *n* = 2), American Diabetes Association (ADA; *n* = 2), National Diabetes Data Group (NDDG; *n* = 2), O’Sullivan and Mahan (*n* = 1), Spanish Group for Diabetes (*n* = 1), Australasian Diabetes in Pregnancy Society (ADIPS; *n* = 2) and World Health Organization 1998/1999 (*n* = 2). Five studies used institution-based diagnostic criteria, while three studies did not report which diagnostic criteria were used. Pregestational diabetes was determined through hospital records and/or by the medication taken by patients. The studies were conducted in different populations, which included: Omani, Saudi, African, Non-Hispanic black, Australian, Asian, Middle Eastern, Indian, Caucasian and Hispanic. Many of the studies were retrospective and did not report the time of assessment of pregnancy outcomes. Twelve studies included in this review defined one or more of the adverse outcomes; however, definitions and/or cut-offs varied across studies, while eight studies did not define outcomes.

Congenital anomalies included cardiovascular, central nervous system, cleft lip and palate, trisomy 21, gastrointestinal, musculoskeletal, and urogenital anomalies/malformations and were referred to differently across studies, which included: congenital anomalies/malformations/abnormalities, birth defects, congenital defects, foetal anomalies/malformations, and neonatal deformities. For the purpose of this review, these were collectively referred to these as congenital anomalies. Moreover, the majority (92.31%) of the studies that reported on congenital anomalies reported the overall incidence and not the incidence of the individual congenital anomalies in their comparisons. Due to significant heterogeneity between studies and the low-quality assessment scores for a few studies, a meta-analysis was not performed, as this may lead to an inaccurate estimate of overall effect size [53].

### 3.3. Quality Assessment of Included Studies

The quality of the 20 studies included in this review ranged from unsatisfactory to very good with scores ranging from 4 to 7 and an average score of 5.5. Three studies scored unsatisfactory (4), seven studies scored fair (5), six studies scored good (6), and four studies scored very good (7) (Appendix A). The studies that rated good and very good were due to controlling for confounding factors, while studies that rated fair and unsatisfactory were affected by not controlling for confounders. The majority of the studies included in this review were retrospective and, therefore, were not able to control for confounders. Due to the narrative nature of this review, all studies were included for analysis despite their risk of bias rating.

### 3.4. Qualitative Synthesis

Of the nine studies that compared combined data for pregestational T1DM and T2DM combined with GDM, the most common adverse outcome reported was C/S (*n* = 7), followed by preterm birth (*n* = 7), congenital anomalies (*n* = 7), pre-eclampsia (*n* = 6), neonatal hypoglycaemia (*n* = 5), macrosomia (*n* = 4), NICU admission (*n* = 4), stillbirth (*n* = 4), Apgar score (*n* = 4), LGA (*n* = 3) RDS (*n* = 3) and IOL (*n* = 2). Of the eleven studies that separately compared pregestational T1DM and T2DM with GDM, the most common adverse outcome reported was C/S (*n* = 10), followed by preterm birth (*n* = 7), macrosomia (*n* = 7), congenital anomalies (*n* = 6), pre-eclampsia (*n* = 4), stillbirth (*n* = 4), neonatal hypoglycaemia (*n* = 3), IOL (*n* = 3), Apgar score (*n* = 3), LGA (*n* = 3), miscarriage (*n* = 2), NICU (*n* = 2) and RDS (*n* = 1). Certain studies subdivided GDM into true GDM (fasting glucose < 7 mmol/L and oral glucose tolerance test (OGTT) 2 h < 11.1 mmol/L) and overt GDM (fasting glucose ≥ 7 mmol/L or OGTT 2 h ≥ 11.1 mmol/L). For the purpose of this review, we focused on outcomes for true GDM.

C/S, preterm birth, and congenital anomalies were the most reported adverse outcomes, while the least reported outcomes were IOL, RDS and miscarriage. Other adverse outcomes reported included preeclampsia, neonatal hypoglycaemia, macrosomia, NICU admissions, stillbirths, LGA and Apgar scores. The majority of the adverse outcomes were higher in pregestational T1DM and T2DM compared to GDM. However, there were a few adverse outcomes that were more common in GDM compared to pregestational T1DM and/or T2DM (Table 1).

*Caesarean section (C/S)*. Of the studies that compared pregestational diabetes (combined T1DM and T2DM) with GDM, four studies reported higher rates of C/S in pregestational diabetes compared to GDM [20,27,43,46], while similar rates were reported in two studies [25,44]. Hyari et al., 2013, reported slightly higher rates of C/S in women with GDM compared to pregestational diabetes [22]. Of the studies that compared pregestational T1DM and T2DM separately with GDM, six studies reported higher rates of C/S in T1DM and T2DM compared to GDM [17,21,48,50,51,52]. Al-Nemri reported higher rates of elective C/S in pregestational T1DM (25.0%) and T2DM (34.3%) compared to GDM (15.7%), but similar rates for emergency C/S [14]. Petticca et al., 2009, reported higher rates of C/S in pregestational T1DM (51.6%) compared to pregestational T2DM (38.0%) and GDM (38.0%), with the latter diabetes types showing similar rates of C/S [24]. Soepnel et al., 2018, reported higher rates of C/S in pregestational T2DM (78.4%) compared to T1DM (67.1%) and GDM (67.8%), with the latter showing similar rates [15]. In contrast, Huddle et al., 1993, reported a higher rate of C/S in GDM (56.0%) compared to pregestational T1DM (39.8%), but similar rates in GDM compared to pregestational T2DM (55.5%) [47]. Taken together, these results demonstrate that C/S is more common in women with pregestational T1DM and T2DM than in women with GDM.

*Preterm birth*. Of the studies that compared pregestational diabetes (combined T1DM and T2DM) with GDM, five studies reported higher rates of preterm birth in pregestational diabetes compared to GDM [22,43,44,45,46], while two studies reported higher rates in GDM compared to pregestational diabetes [20,27]. Of the studies that compared pregestational T1DM and T2DM separately with GDM, six studies reported higher rates of preterm birth in pregestational T1DM and T2DM compared to GDM [17,21,24,50,51,52]. Stogianni et al., 2019, reported higher rates of preterm birth in pregestational T2DM (46.0%) compared to pregestational T1DM (35.0%) and GDM (12.0%), and higher rates in pregestational T1DM compared to GDM [48]. These results show that preterm birth is more common in women with pregestational T1DM and T2DM than in women with GDM.

*Congenital anomalies*. Higher rates of congenital anomalies were reported in pregestational diabetes (combined T1DM and T2DM) compared to GDM in four studies [16,22,25,27], while Barakat et al., 2010, reported higher rates in GDM (8.9%) compared to pregestational diabetes (5.6%) [20]. In contrast, two studies reported no significant difference in the rates of congenital anomalies between pregestational diabetes and GDM [43,45]. When comparing T1DM and T2DM separately with GDM, four studies reported higher rates of congenital anomalies in pregestational T1DM and T2DM compared to GDM [14,17,24,47]. Of these, two reported higher rates of congenital anomalies in pregestational T2DM compared to pregestational T1DM and GDM, and higher rates in pregestational T1DM compared to GDM [14,47]. In contrast, two studies reported no significant difference in rates of congenital anomalies between the three diabetic groups [15,21]. Although discrepant results are reported, the majority of studies showed that congenital anomalies are more common in neonates born to mothers with pregestational T1DM and T2DM than in neonates born to mothers with GDM.

*Pre-eclampsia*. Higher rates of pre-eclampsia were reported in pregestational diabetes (combined T1DM and T2DM) compared to GDM in three studies [43,45,46], while two studies reported higher rates in GDM compared to pregestational diabetes [22,27]. El Mallah et al., 1997, reported no difference in the rates of pre-eclampsia between pregestational diabetes (1.4%) and GDM (2.0%) [44]. Pre-eclampsia was also compared in pregnant women with pregestational T1DM and T2DM separately with GDM. Higher rates of pre-eclampsia were reported in pregestational T1DM compared to T2DM and GDM in three studies, with the latter occurring at similar rates [17,24,50]. Soepnel et al., 2019, reported no significant difference in the rates of pre-eclampsia across the three diabetic groups [15]. Taken together, pre-eclampsia is more common in women with pregestational T1DM and T2DM than GDM and more common in pregestational T1DM.

*Neonatal hypoglycaemia*. Three studies reported higher rates of neonatal hypoglycaemia in pregestational diabetes (combined T1DM and T2DM) compared to GDM [27,43,46], while two studies reported no difference in the rates of neonatal hypoglycaemia between pregestational diabetes and GDM [22,44]. When comparing neonatal hypoglycaemia between T1DM and T2DM separately with GDM, Yamamoto et al., 2020, reported higher rates in T1DM (27.5%) and T2DM (18.3%) compared to GDM (5.0%) [52] and Huddle at al., 1993, reported higher rates of neonatal hypoglycaemia in neonates born to mothers with pregestational T1DM (4.2%) and GDM (4.2%) compared to neonates born to mothers with pregestational T2DM (3.6%) [47]. However, Al-Nemri et al., 2018, reported no difference in the rates of neonatal hypoglycaemia across the three diabetic groups [14]. These results show that rates of neonatal hypoglycaemia are more common in neonates born to mothers with pregestational T1DM and T2DM compared to neonates born to mothers with GDM.

*Macrosomia*. Higher rates of macrosomia were reported in pregestational diabetes (combined T1DM and T2DM) compared to GDM in three studies) [20,22,44], while Abu-Heija et al., 2015, reported no significant difference in the rates of macrosomia between pregestational diabetes (10.3%) and GDM (4.9%) [43]. Macrosomia was also reported when comparing T1DM and T2DM separately with GDM. Two studies reported higher rates in T1DM and T2DM compared to GDM [15,50]. Peticca et al., 2009, reported higher rates of macrosomia in T1DM (17.2%) and GDM (12.2%) compared to T2DM (11.1%) [24], while Van Zyl and Levitt reported higher rates of macrosomia in GDM (9.2%) compared to pregestational T1DM (8.5%) and T2DM (8.2%) [17]. However, three studies reported no significant difference in the rates of macrosomia between the three diabetic groups [14,21,48]. Altogether, these studies indicate that macrosomia is more common in neonates born to mothers with pregestational diabetes T1DM and T2DM compared to GDM.

*NICU admissions*. When NICU admissions were compared between pregestational diabetes (combined T1DM and T2DM) and GDM, four studies reported higher rates of NICU admissions in pregestational diabetes compared to GDM [20,43,45,46]. NICU admissions were also reported when comparing T1DM and T2DM separately with GDM. Yamatoto et al., 2020, reported higher rates of NICU admissions in T1DM (55.5%) and T2DM (31.0%) compared to GDM (14.0%) [52], while A-Nemri et al., 2018, reported higher rates of NICU admissions in pregestational T1DM (66.7%) compared to pregestational T2DM (16.0%) and GDM (10.2%), with the latter showing similar rates [14]. These results demonstrate that NICU admissions are more common in neonates born to mothers with pregestational diabetes T1DM and T2DM compared to neonates born to mothers with GDM.

*Stillbirth.* When stillbirth was compared between pregestational diabetes (combined T1DM and T2DM) and GDM, higher rates of stillbirth were reported in pregestational diabetes compared to GDM in two studies [44,46]. However, two studies reported no difference in the rates of stillbirths between pregestational diabetes and GDM [20,45]. When comparing T1DM and T2DM separately with GDM, higher rates of stillbirths were reported in pregestational T1DM and T2DM compared to GDM in three studies [17,24,49], while Huddle et al., 1993, reported higher rates in T2DM (4.7%) compared to T1DM (3.3%) and GDM (4.0%) with the latter occurring at a similar rate [47]. Altogether, these results demonstrate that stillbirths are more common in neonates born to mothers with pregestational T1DM and T2DM compared to neonates born to mothers with GDM.

*Apgar score*. Low Apgar scores (<7) were compared between pregestational diabetes (combined T1DM and T2DM) and GDM. Barakat et al., 2010, reported higher rates of low Apgar scores in pregestational diabetes (24.1%) compared to GDM (22.1%) [20], while three studies reported no difference in the rates of low Apgar scores between pregestational diabetes and GDM [43,44,46]. Low Apgar scores were also reported when comparing T1DM and T2DM separately with GDM. Gualdani et al., 2021, reported lower Apgar scores in T1DM (5.4%) compared to T2DM (2.5%) and GDM (1.3%) [21], while two studies reported similar rates of low Apgar scores in T1DM and T2DM, although higher than GDM [24,48]. These findings indicate that low Apgar scores present at a similar rate in neonates across the three diabetic groups.

*Large for gestational age (LGA)*. Two studies reported higher rates of LGA in GDM compared to pregestational diabetes (combined T1DM and T2DM) [25,27], while Shand et al., reported higher rates of LGA in pregestational diabetes (35.0%) compared to GDM (15.9%) [46]. LGA was also reported when comparing T1DM and T2DM separately with GDM. Two studies reported higher rates of LGA in T1DM and T2DM compared to GDM [48,52]. In contrast, Gualdani et al., 2021, reported no significant difference between the three diabetic groups [21]. Altogether, the results show that LGA is more common in neonates born to mothers with pregestational T1DM and T2DM compared to neonates born to mothers with GDM.

*Induction of labour (IOL)*. Two studies reported no difference in the rates of IOL between pregestational diabetes and GDM [43,46]. In the comparison of T1DM and T2DM separately with GDM, López-de-Andrés et al., 2020, reported higher rates of IOL in pregestational T1DM (29.6%) and T2DM (30.4%) compared to GDM (22.6%) [51], while Peticca et al., 2009, reported higher rates of IOL in T1DM (44.7%) and GDM (38.3%) compared to T2DM (36.6%) [24]. In contrast, Van Zyl and Levitt, 2018, reported higher rates of IOL in GDM (30.0%) compared to T1DM (11.8%) and T2DM (18.6%) [17]. These results show that IOL occurs at similar rates in women with pregestational T1DM and T2DM and GDM.

*Respiratory distress syndrome (RDS)*. When comparing pregestational diabetes (combined T1DM and T2DM) and GDM, higher rates of RDS were reported in pregestational diabetes compared to GDM in two studies [27,43], while Barakat et al. reported higher rates in GDM (2.8%) compared to pregestational diabetes (1.6%) [20]. In the comparison of T1DM and T2DM separately with GDM, Al-Nemri et al. reported higher rates of RDS in T1DM (44.4%) compared to T2DM (13.9%) and GDM (13.5%) with similar rates occurring in the latter [14]. These results demonstrate that RDS is more common in neonates born to mothers with pregestational T1DM and T2DM than in neonates born to mothers with GDM.

*Miscarriage*. When comparing T1DM and T2DM separately with GDM, higher rates of miscarriage were reported in T1DM compared to T2DM and GDM in two studies [15,17]. These results indicate that miscarriages are more common during pregestational T1DM compared to pregestational T2DM and GDM.

## 4. Discussion

Adverse outcomes associated with maternal diabetes are reported to be more common in women with pregestational diabetes compared to GDM; although, conflicting results have been reported [14,15,16,17,27,47,48]. In this systematic review, we summarise and synthesise studies that have compared adverse pregnancy outcomes in pregnancies complicated by pregestational diabetes and GDM. Findings from this review confirm that both pregestational diabetes and GDM are associated with pregnancy complications including C/S, preterm birth, congenital anomalies, pre-eclampsia, neonatal hypoglycaemia, macrosomia, NICU admission, stillbirth, Apgar score, LGA, IOL, RDS and miscarriage. Although conflicting results were reported in a few studies, the majority of studies report that adverse outcomes are more common in pregnancies complicated by pregestational diabetes than GDM. This review did not identify studies that compared long-term adverse outcomes in women with pregestational diabetes and GDM.

Thirteen perinatal complications, C/S, preterm birth, congenital anomalies, pre-eclampsia, neonatal hypoglycaemia, macrosomia, NICU admission, stillbirth, Apgar score, LGA, IOL, RDS and miscarriage, which are amongst the most common maternal and foetal adverse outcomes reported in the literature, were compared in this review. C/S was the most common adverse outcome reported. Although it is accepted that not all C/S may be considered an adverse pregnancy outcome [54], it is often recommended by health care providers as a strategy to reduce the risk of perinatal complications associated with maternal diabetes [55,56]. Preterm birth is defined as birth before 37 completed weeks of gestation [33] and is the leading cause of mortality in children younger than five years. Infants who survive preterm birth often present with poor neurodevelopment and cognitive disabilities [57] and behavioural and emotional difficulties [58]. Congenital anomalies, which refer to structural or functional malformations that occur during intrauterine life, are associated with hyperglycaemia during the period of organogenesis that occurs in the first trimester of pregnancy. Maternal hyperglycaemia leads to the increased production of reactive oxygen species (ROS), resulting in DNA and membrane damage and the subsequent induction of apoptosis, causing malformations in major organs of the developing foetus [23]. Pre-eclampsia is characterised by hypertension, which usually develops after 20 weeks of gestation [34] and is considered the leading cause of maternal morbidity and mortality among women who have diabetes [59]. The condition is thought to occur due to endothelial dysfunction, dyslipidaemia, and inflammation associated with diabetes [60,61].

Macrosomia refers to giving birth to babies weighing more than 4 kg and is considered the most common adverse outcome associated with maternal diabetes [6,29]. The condition is thought to occur due to increased placental transport of glucose and other nutrients from the mother to the foetus, resulting in accelerated growth [7,62]. Macrosomia is associated with several complications including, neonatal hypoglycaemia and premature birth [55,56]. Abnormal placental supply of nutrients results in abnormal foetal growth, including foetal growth restriction (FGR) and foetal overgrowth, and is associated with increased neonatal mortality. LGA refers to a foetus that weighs in >90th percentile of the birth chart [36]. LGA is associated with an increased rate of C/S and neonatal hypoglycaemia, including a longer hospital stay in mothers with diabetes [63,64]. Neonatal hypoglycaemia is defined as a plasma glucose value < 1.65 mmol/L in the first 24 h of life and <2.5 mmol/L onwards [37]. Hypoglycaemia in neonates occurs due to continuous placental transport of glucose and other nutrients from the mother to the foetus, which results in hyperinsulinaemia, which leads to a fall in glucose levels during and post-delivery [65,66]. Hyperinsulinism is very common in infants of mothers with diabetes [37]. Hyperinsulinaemia in the foetus may also lead to RDS at birth. RDS is defined by the need to supplement neonatal oxygen to maintain a saturation of over 85% within the first 24 h after birth and also radiological features [42]. The development of RDS has been attributed to the inhibitory effects of insulin on the expression of surfactant proteins A and B in lung epithelial cells, resulting in decreased production of surfactants and delayed pulmonary maturation [28,42,67].

Placental abnormalities and congenital malformations are major risk factors for stillbirth and neonatal death, which represent the extreme end of the spectrum of complications in diabetic pregnancies [49]. Stillbirth is defined as the death of a foetus at ≥22 weeks of gestation or birth weight of ≥500 g [35]. Unexplained stillbirths at term in maternal diabetes are attributed to maternal hyperglycaemia and foetal hyperinsulinaemia, foetal hypoxia and acidaemia and cardiomyopathy due to glycogen deposition in the myocardium [68,69]. Maternal diabetes has also been associated with an increased risk of miscarriages and habitual abortions [70,71]. Animal models have shown that maternal diabetes affects pre-implantation in the embryo developmental stages. In vivo and in vitro studies show that hyperglycaemia leads to an overexpression of *Bax,* (Bcl-2-associated X), which is a death-promoting protein associated with increased apoptotic morphological changes and is reversed by insulin [72]. In women with diabetes, IOL is recommended to minimise birth complications associated with macrosomia and the risk for stillbirth [73]. A Cochrane review by Boulvain et al., 2001, showed that induction of labour lowered the prevalence of macrosomia without increasing the risk of caesarean section [74].

Furthermore, poor glucose control in the third trimester may lead to perinatal asphyxia and low Apgar scores [75,76]. Apgar score is a clinical method used to assess the wellbeing of a neonate at 1 min and 5 min after birth. The Apgar score assesses elements such as skin colour/tone, heart rate, reflexes, muscle tone and respiration [41]. Apgar scores may predict long-term neurological disabilities in infants [77,78]. Foetal complications are associated with increased admissions to the neonatal intensive care unit (NICU), which is therefore often used as an indicator of adverse pregnancy outcomes [77,78].

Limitations of the studies included in this review may hinder our ability to draw significant conclusions. There was heterogeneity across studies in terms of population characteristics, the diagnostic criteria used, the definitions used for pregnancy outcomes (e.g., preterm birth, Apgar scores) and different medication regimens (diet, metformin, and insulin). It has been widely reported that ethnicity [79,80], advanced maternal age [81], diet [82], socioeconomic status [83] and medication regimen [48] influence pregnancy outcomes. Furthermore, the majority of studies were retrospective and were dependent on the accuracy of medical records and databases, which may negatively affect study accuracy [84]. Many of the included studies had a poor risk of bias scores, which were mainly affected by the lack of accounting for confounding factors, which may have affected the accuracy of study findings. Excluding studies with unsatisfactory ratings from the analysis, did not affect the overall conclusions of the review, and similar to studies with a satisfactory and high risk of bias scores, showed that adverse outcomes were more common in pregestational T1DM and T2DM compared to GDM. Therefore, all the studies were included as the data were deemed valuable for the purpose of this narrative review.

Despite the inconclusive results from this review, it is evident that pregestational diabetes poses a greater risk for pregnancy complications than GDM and emphasises the importance of maintaining optimal glucose control during the preconception period. Maternal metabolic factors may program physiological adaptation to pregnancy, thereby affecting pregnancy outcomes [85,86]. The importance of preconception health is increasingly acknowledged as a key determinant of pregnancy success, with increasing attention shifting to preconception intervention [86]. A population-based study in Canada reported that a 10% weight reduction in the preconception period decreased the risk of developing GDM, pre-eclampsia, preterm delivery, macrosomia and stillbirth [87]. Another study showed that women who underwent bariatric surgery prior to conception had a lower risk of developing GDM, hypertensive disorders and macrosomia [88]. Furthermore, increased physical activity before conception is associated with a lower risk of GDM [89] and pre-eclampsia [90]. Taken together, these studies demonstrate a strong relationship between preconception health and pregnancy outcomes. The mechanisms that underlie these links are not known, but are likely to involve an array of genetic, epigenetic and environmental factors that interact to affect physiological adaptation during pregnancy.

While acknowledging the importance of preconception health and optimal glucose control during pregnancy, the importance of GDM prevention should not be underestimated. As with pregestational diabetes, albeit less common, GDM was also associated with several adverse pregnancy outcomes. Importantly, these complications can be avoided by preventing the development of GDM. During pregnancy, lifestyle modifications that include diet and physical activity have been shown to prevent GDM [89,91,92,93]. Although not addressed in this review, recent studies have highlighted the occurrence of early-onset GDM, defined as GDM that can be detected in women before 24 weeks of gestation [94]. These women have an increased risk of adverse pregnancy outcomes compared to women with “normal” GDM diagnosed at 24–26 weeks [95,96], and highlights the need to diagnose early pregnancy glycaemia as recently reported by McIntyre et al. [97].

## 5. Future Perspectives

The majority of studies included in this review were retrospective. In addition, we did not identify articles that investigated long-term adverse outcomes in women with pregestational T1DM and T2DM, and GDM. Therefore, there is a need for prospective, longitudinal studies in the future to more accurately compare short- and long-term adverse pregnancy outcomes across diabetes types. Preterm birth was one of the most common adverse outcomes reported in this review. The optimal timing of delivery for women with pregestational diabetes is not known due to a lack of published trials [98]; therefore, there is a need for more studies to determine the optimal time to deliver babies born to mothers with diabetes as this will reduce the complications associated with preterm delivery.

## 6. Conclusions

In conclusion, the findings from this review confirm that adverse pregnancy outcomes are more common in women with pregestational diabetes compared to women with GDM. These findings highlight the importance of preconception health and the need to educate women of reproductive age who have diabetes or who are at risk of diabetes about the importance of pre-pregnancy care and maintaining good glycaemic control to improve pregnancy health and reduce the risk of adverse pregnancy outcomes. Another important finding of the review is the high rates of adverse outcomes observed in women with GDM, and the need for intervention strategies to prevent the development of GDM.

## Figures and Tables

**Figure 1 ijerph-19-10846-f001:**
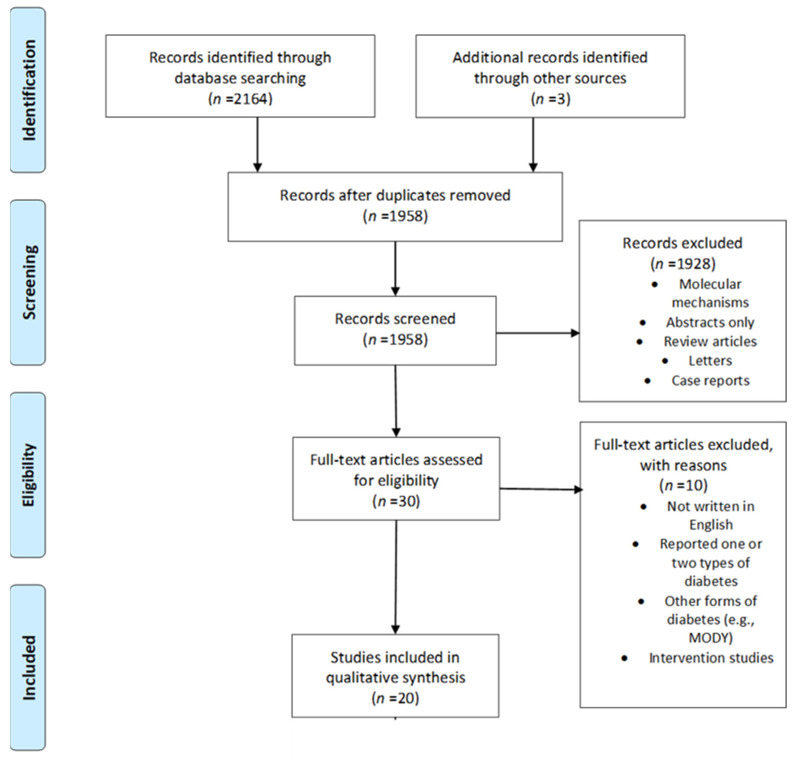
Flow diagram for the search criteria; MODY—maturity-onset diabetes of the young.

**Table 1 ijerph-19-10846-t001:** The frequency of adverse pregnancy outcomes.

Adverse Outcome	Increased in Pregestational Diabetes	Increased in GDM	No Difference
Caesarean section	[15,17,20,21,24,27,43,46,48,50,51,52]	[22,47]	[25,44]
Preterm birth	[17,21,22,24,43,44,45,46,48,50,51,52]	[20,27]	
Congenital anomalies	[14,16,17,22,24,25,27,47]	[20]	[15,21,43,45]
Pre-eclampsia	[17,24,43,45,46,50]	[22,27]	[15,44]
Neonatal hypoglycaemia	[27,43,46,47,52]		[14,22,44]
Macrosomia	[15,20,22,24,44,50]	[17]	[14,21,43,48]
NICU admission	[14,20,43,45,46,52]		
Stillbirth	[17,24,44,46,47,49]		[20,45]
Apgar score	[20,21,24,48]		[43,44,46]
Large for gestational age	[46,48,52]	[25,27]	[21]
Induction of labour	[24,43,51]	[17]	[46]
Respiratory distress syndrome	[14,27,43]	[20]	
Miscarriage	[15,17]		

## Data Availability

Not applicable.

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
