# Peer review of "A Systematic Review to Compare Adverse Pregnancy Outcomes in Women with Pregestational Diabetes and Gestational Diabetes"

_ijerph, 2022, doi:10.3390/ijerph191710846_

Round 1
Reviewer 1 Report
General comment
The aim of this study was to aims to summarize and synthesize studies that have compared adverse pregnancy outcomes in pregnancies complicated by pregestational diabetes and GDM. This study found that pregestational diabetes is associated with more frequent pregnancy complications than GDM. Overall, the topic is important. But I don’t think the author has done a systematic and scientific comparison of the association between different gestational diabetes subtypes and adverse pregnancy outcomes, and the conclusion is not credible. The author should conduct further meta-analysis to prove the potential association. I don’t think the current article can be published.
Specific comments
1.Line 109-111, did animal the author delete animals’ studies and molecular mechanism studies and duplicate studies?
2.Line 127, the Newcastle-Ottawa Scale was used to evaluate the study quality of case-control study and cohort study. Are all the studies included case-control study and cohort study? Please add description of selection of literature quality assessment scales.
3. The authors did not collect the time of diagnosis of gestational diabetes and the time of adverse pregnancy outcomes. How to determine the chronological order of diabetes and adverse pregnancy outcomes?
4. The authors did not collect the diagnosis time of gestational diabetes and adverse pregnancy outcomes. How to determine the chronological order of diabetes and adverse pregnancy outcomes?
5.Figure 1: The process of excluding articles was not detailed, please complete the process of excluding records.
6.Line 175, What did the C/S mean? Please use the full name and abbreviation if the word was first appearing for the first time.
7. Line 222, How were congenital abnormalities defined? If the author was referring to birth defects, there were many subtypes of birth defects, and such comparisons were meaningless and too crude.
8. It is recommended to add definitions and diagnostic criteria for each adverse pregnancy outcome in the Methods section.
9. When the authors compared the results of different studies, did the studies use the same diagnostic criteria? Was the study population the same? The author should specifically state this, this question is important.
10. Overall, the authors can perform meta-analyses to explore potential associations, the authors' current review does not explain and illustrate the association of different subtypes of gestational diabetes with adverse pregnancy outcomes while the conclusions are not credible.
Reviewer 2 Report
The REVIEW of the study by Nompumelelo Malaza, Matladi Masete, Sumaiya Adam, Stephanie Dias , Thembeka Nyawoand, Carmen Pheiffer: A systematic review to explore the association between maternal diabetes and adverse pregnancy outcomes.
This is a well written and comprehensive review that addresses the consequences of various types of diabetes mellitus (i.e. type 1, type 2 and gestational diabetes – GDM) on various pregnancy outcomes. The authors conclude that adverse outcomes are more common in pregnancies complicated by pregestational diabetes than GDM. Though this is well researched review, that addresses all important outcomes, in my opinion there is an aspect that is missing and should be addressed at least in the discussion.
In particular, I mean the issue of timing for screening for GDM. This is typically done in the late second and early third trimester, but a significant number of women have an “early GDM”, e.g. present already in the first trimester, and often not diagnosed. For instance in the study performed entirely on a cohort of healthy Caucasian women [Lewandowski K et al. High prevalence of early (1st trimester) gestational diabetes mellitus in Polish women is accompanied by marked insulin resistance — comparison to PCOS model. Endokrynol Pol. 2022;73(1):1-7. doi: 10.5603/EP.a2021.0095.], prevalence of early GDM was between 9% and 14%, depending on the criteria used (i.e. WHO versus IADPSG). Also women with the “early GDM” were shown to be significantly more insulin resistant. Hence, at least in theory, there is a possibility that women with the “early GDM” might be more prone to pregnancy-related complications. In such setting, in terms of pregnancy-related risks this would place them closer to women with pregestational diabetes. Even, if answers to such questions are currently not available, this controversy should be at least highlighted, particularly in view the IADPSG statement that “Normative data regarding early pregnancy glycaemia and consequences of its detection and treatment are urgently required and should be a priority for future research” [McIntyre HD, et al. Issues With the Diagnosis and Classification of Hyperglycemia in Early Pregnancy. Diabetes Care. 2016; 39(1): 53–54, doi: 10.2337/dc15-1887].
Reviewer 3 Report
In this systematic review, the authors review pregnancy outcomes of women with pregestational diabetes (T1 or T2) compared with outcomes from women diagnosed with gestational diabetes (GDM). Overall this is a well written and informative review but could be strengthened with the following changes:
1. The title is misleading. Specifically, the authors compare pregestational diabetes to a GDM diagnosis. This should be reflected in the title.
2. In Methods section 2.2, inclusion and exclusion criteria, the authors state they were answering the following “questions” and then go on to only state one question. They should remove the title “Question 1” given that there is only one question being addressed in this review.
3. Inclusion of a second table would be helpful. In this table the outcomes investigated would be the first column (i.e. C/S, macrosomia etc). A second column could list the articles that reported an increased risk for women with pregestational diabetes compared with GDM and a third column could list the articles that reported no difference between the groups. Given that this is the crux of the paper, a better way to present this information would help the reader, rather than the text writing out each individual outcome and listing the articles.
4. The paragraph in the discussion that extends from line 93 through line 156 is rather long and contains information outside the scope of this review. It should be heavily edited or removed completely.
5. There is a disconnect between Supplementary Table S2 in the Excel file and the reference to this Supplementary Table in the manuscript (line 130 in the Methods section). The authors state that this table should report on a scoring system, and it is a blank table with no information on scoring.
Round 2
Reviewer 1 Report
As the author said, there was significant heterogeneity across studies, but this could be solved and discussed through statistical analysis such as subgroup analysis. The author did not answer my question well but avoided the possibility of conducting meta-analysis , I still think this study has no research significance.
Author Response
Dear Reviewer
Thank you for the comment and we acknowledge the importance of the comment. Due the heterogeneity of the studies included, a meta-analysis and subgroup analysis were not performed. We do have certain studies that rated low in the risk of bias analysis, however, all the studies were included as the data were deemed valuable for the purpose of this narrative review.
Thank you